# Testing the Feasibility and Dietary Impact of Macaroni Fortified with Green Tea and Turmeric Curcumin Extract in Diabetic Rats

**DOI:** 10.3390/foods12030534

**Published:** 2023-01-25

**Authors:** Nuntouchaporn Hutachok, Pimpisid Koonyosying, Narisara Paradee, Rajnibhas Sukeaw Samakradhamrongthai, Niramon Utama-ang, Somdet Srichairatanakool

**Affiliations:** 1Oxidative Stress Cluster, Department of Biochemistry, Faculty of Medicine, Chiang Mai University, Chiang Mai 50200, Thailand; 2Cluster of High-Value Products from Thai Rice and Plant for Health, Chiang Mai University, Chiang Mai 50100, Thailand; 3Cluster of Innovative Food and Agro-Industry, Chiang Mai University, Chiang Mai 50100, Thailand; 4Division of Product Development Technology, Faculty of Agro-Industry, Chiang Mai University, Chiang Mai 50100, Thailand

**Keywords:** green tea, turmeric curcumin, macaroni, satisfaction, glucose, triglyceride, cholesterol

## Abstract

Macaroni is a commercially available Italian food product that is popular among consumers around the world. The supplementation of green tea extract (GTE) and turmeric curcumin extract (TCE) in macaroni may serve as promising and beneficial bioactive ingredients. We aimed to produce functional macaroni, assess the degree of consumer satisfaction and study the antidiabetic activity in diabetic rats. In this study, macaroni was fortified with GTE, TCE and a mixture of GTE and TCE ratio of 1:1, *w*/*w* (GTE/TCE). The resulting products were then analyzed in terms of their chemical compositions, while the degree of consumer satisfaction was monitored and the hypoglycemic and hypolipidemic effects in streptozotocin (STZ)-rats were investigated. GTE/TCE-M exhibited the strongest antioxidant activity (*p* < 0.05), while phenolics were most abundant in GTE-M. The overall preference for GTE-M, TCE-M and GTE/TCE-M were within ranges of 4.7–5.1, 5.9–6.7 and 6.2–8.2, respectively, in the nine-point hedonic scale. Consumption of these three preparations of macaroni (30 and 300 mg/kg each) neither decreased nor exacerbated increasing blood glucose levels in diabetic rats, while GTE-M (30 mg/kg) tended to lower increased serum triglyceride and cholesterol levels. In conclusion, GTE/TCE-M containing high amounts of bioactive EGCG and curcumin exerted the strongest degree of antioxidant activity and received the highest level of acceptance. Importantly, consumption of GTE-M tentatively ameliorated serum lipid abnormalities in diabetic STZ-induced rats by inhibiting lipase digestion and lipid absorption. Herein, we are proposing that GTE-fortified macaroni is a functional food that can mitigate certain metabolic syndromes.

## 1. Introduction

Plants synthesize secondary metabolites, such as polyphenolics, flavonoids, stilbenes, and curcuminoids, thereby presenting antioxidant, reactive oxygen species (ROS)-scavenging, anti-lipid peroxidation, antidiabetic and anti-carcinogenesis properties [1]. Statistically, tea is the most popular and widely consumed beverage in the world. Green tea and black tea are produced from the common tea plant (*Camellia sinensis*) in many regions of the world. Green tea extract (GTE) contains high amounts of catechin (C), epicatechin (EC), epicatechin 3-gallate (ECG), epigallocatechin (EGC), gallocatechin 3-gallate (GCG), and epigallocatechin 3-gallate (EGCG), of which EGCG is the most abundant, while ECG exhibited the highest degree of antioxidant activity [2,3,4]. Interestingly, GTE attenuated hepatic health through antioxidation, anti-inflammation, and anti-steatotic effects in obese mice [5]. In addition, green tea catechins have been reported to manipulate lipid metabolism [6,7,8], inhibit liver gluconeogenesis [9], decrease hyperglycemia and decrease oxidative stress in diabetic rats [10,11]. Likewise, white and green tea treatments were found to decrease the uptakes of glucose and cholesterol in human hepatocellular carcinoma (HepG2) cells [12]. Moreover, GTE-supplemented food decreased serum and liver triglyceride (TG) levels and hepatic fatty acid synthase (FAS) activity. This would suggest a hypotriglyceridemic effect in rats [13]. Turmeric and curry powders derived from the *Curcuma longa* rhizome have long been used as spices and seasoning products in Middle Eastern and Asian countries. Importantly, turmeric curcumin extract (TCE) is known to contain curcuminoids, specifically curcumin, desmethoxycurcumin and bis-desmethoxycurcumin [14,15]. TCE is also known to deliver certain health benefits that are related to its potent antioxidant, iron-chelating, prebiotic, anti-inflammatory, hypotensive, and cognitive-improvement properties [16,17,18,19,20,21]. 

Biochemically, ROS are produced not only by mitochondrial electron transport and neutrophil functions, but also by nonenzymatic iron catalysis via Haber-Weiss and Fenton reactions. Excessive ROS contents can be harmful to certain biomolecules including polyunsaturated fatty acids, nucleic acids and proteins in living cells and organisms [22]. Accordingly, cellular and systemic antioxidant defense mechanisms are raised to scavenge excessive ROS levels, prevent oxidative tissue damage, reduce inflammation, and treat metabolic syndrome and cardiovascular disease (CVD) [23,24]. Excessive lipid accumulation and oxidative stress in the liver can cause nonalcoholic fatty liver disease (NAFLD). Nowadays, plant extracts and their constituents are being processed or formulated to mitigate ROS-related diseases, as well as certain non-communicable diseases (NCD) and metabolic syndromes, especially, diabetic mellitus (DM), hypertension and hyperlipidemias. Likely, polyphenolics, or curcuminoids per se, mixed into food could help to prevent pathogenesis and relieve NCD, consequently resulting in elevated states of health of subjects, improving their general well-being and longevity, and offering them a better quality of life. 

At present, turmeric and green tea are widely used in the food industry and in the cosmetic sciences. They are also being used to develop supportive treatments and alternative medications. For instance, it has been reported that certain dietary cocktails can increase a patient’s level of antioxidant defense, which may provide preventive effects at certain stages of specific diseases [25]. Surprisingly, GTE and TCE-formulated food restored increased plasma levels of redox-active iron and lipid-peroxidation products in transfusion-dependent β-thalassemia patients with iron overload [26]. Additionally, curcuminoids formulated in the GPO cocktails diminished serum levels of redox-active iron and oxidative stress parameters in β-thalassemia/HbE patients [21]. Collectively, research on the effects of green tea and turmeric curcumin in health and diseases has confirmed their antioxidant, anti-inflammatory, antidiabetic, and lipid-lowering activities. 

Macaroni is one of the most popular dishes across the world and offers a range of health benefits to consumers. Throughout the world, macaroni is consumed at breakfast, in lunch buffets and at family dinners. It is a versatile food that is loved by all ages of people worldwide. Azolla fern powder and its alginate encapsulated form were tested for suitability in the fortification of macaroni and received high acceptability index scores [27]. Unfortunately, consumption of certain spicy, pickled, highly-processed and high-fat foods, such as sausage and bologna, pickles, soft drinks, grains, tea, salt, pizza, watermelon, pepper, and macaroni may cause dyspepsia [28]. In addition, the control and satiating capabilities of our prepared macaroni products were previously investigated in terms of their degree of appreciation and their effects on non-obese men [29]. Moreover, macaroni can be modified by the addition of herb and spice blends, as well as vegetables, to improve the flavor and increase the intake of vegetables among consumers [30]. To date, there have not yet been any reports on the incorporation of green tea and/or turmeric curcumin in foods especially fresh macaroni. It has been hypothesized that the characteristics and biological activities of GTE and TCE formulated in our macaroni food could attenuate events leading to DM and/or NAFLD. In this study, we prepared macaroni products supplemented with green tea and turmeric curcumin extracts, determined their chemical compositions and antioxidant activity, evaluated the degree of consumer satisfaction, and investigated their anti-diabetic and anti-hyperlipidemic effects on streptozotocin-induced and high-fat diet (HFD)-fed rats.

## 2. Materials and Methods

### 2.1. Chemicals and Reagents

Accordingly, 2,2′-azino-bis(3-ethylbenzthiazoline-6-sulphonic acid (ABTS), 3-(4,5-dimethylthiazol-2-yl)-2,5-diphenyltetrazolium bromide (MTT) reagent, Folin-Ciocalteu reagent, Trolox (6-hydroxy-2,5,7,8-tetramethylchroman-2-carboxylic acid), 2-deoxy-2-({[methyl(nitroso)amino]carbony}amino)-β-D-glucopyranose (streptozotocin, STZ), 5-chloro-2-methoxy-N-[2-(4-sulfamoylphenyl)ethyl]benzamide (glibenclamide), (2S)-2-[(1R)-1-hydroxy-2-[(2S)-5-oxooxolan-2-yl]ethyl]octanoic acid or Orlistat (Xenical^®^, GreifswaldCHEPLAPHARM Arzneimittel GmbH, Germany), and [4-[3-(4-ethylphenyl)butyl]phenyl]-trimethylazanium chloride (cholestyramine resin) were purchased from Sigma-Aldrich Chemicals Company (St. Louis, MO, USA). Subsequently, o-phosphoric acid, acetonitrile and tetrahydrofuran (THF) were purchased from Merck KGaA Group, Darmstadt, Germany. Standard gallic acid (GA) samples were obtained from Sigma-Aldrich Chemicals Company (St. Louis, MO, USA). Semolina durum wheat flour and table salt (Prungtip brand, Thai Refined Salt Company Limited, Bangkok, Thailand), which were food-grade and high-quality products, were purchased from a Yok Grocery Shop located in Chiang Mai Province. 

### 2.2. Preparation of Macaroni Products 

#### 2.2.1. Green Tea Extract-Fortified Macaroni

Fresh tea leaves (*Camellia sinensis*) were acquired from tea fields located in Doi Mon Ngo, Mueangkai Subdistrict, Mae Taeng District, Chiang Mai Province, Thailand. Tea leaves were dried in a microwave oven (Electrolux^®^, Stockholm, Sweden, 20-L capacity, 4000-watts power) at 200 °C for 3–5 min to inactivate polyphenol oxidase. The dried tea leaves were then ground using a blender (Electrolux^®^, Stockholm, Sweden). The powder (15.0 g) was extracted in 80 °C hot deionized (DI) water (100 mL) for 10 min, allowed to cool down, filtered through a clean sheet cloth and collected as a 15% (*w*/*v*) GTE solution. One part of the resulting GTE was used as an additive composition and the other part was dried under a vacuum for 48 h using a freeze-drying machine (LABCONCO™, Labconco Corporation, Kansas City, MO, USA) according to the manufacturer’s instructions. The freeze-dried sample was kept in a brown plastic bottle at −20 °C [26]. Accordingly, green tea extract-enriched macaroni (GTE-M) product was manufactured using the method established by Olfat and colleagues [31] with slight modifications. It was composed of semolina durum wheat flour (64%, *w*/*w*), 15% GTE solution (16.7%. *w*/*w*), GTE powder (2%, *w*/*w*), table salt (1%, *w*/*w*) and clean drinking water (16.3%, *w*/*w*). Afterward, all ingredients were homogenized using a KitchenAid 5K5SS stand mixer, kneaded for 25 min until a dough was formed, chopped into equal sizes and dried in an oven at 75 °C for 3 h using a macaroni-making machine (Marcato, Regina, Italy). Finally, GTE-M was cooked in boiling water (100 ± 2 °C) for 15–20 min and kept in an aluminum-foil bag. 

#### 2.2.2. Turmeric Curcumin Extract-Fortified Macaroni 

Turmeric (*Curcuma longa* Linn.) was purchased from a local market located in Muang District, Chiang Mai Province, Thailand. The turmeric was then rinsed twice with tap water, chopped into small pieces (0.5-mm thick), dried in a vacuum microwave oven (60 °C) for 8–12 h, and ground using a blender (Electrolux^®^, Stockholm, Sweden). The powder (100 g) was then extracted with hot water (100 ± 2 °C) (900 g) for 5 min, cooled down, and filtered through a clean cloth sheet. The resulting 10% (*w*/*v*) turmeric curcumin extract (TCE) food was kept in a dark plastic bottle [26]. Similarly, the turmeric curcumin-enriched macaroni (TCE-M) product was manufactured using the method established by Olfat and colleagues [31] with slight modifications. It was composed of semolina durum wheat flour (69%, *w*/*w*), 10% TCE solution (5%, *w*/*w*), table salt (1%, *w*/*w*) and clean drinking water (25%, *w*/*w*). The ingredients were mixed homogenously, kneaded for 25 min until dough was formed, chopped into equal sizes, and dried in an oven at 75 °C for 3 h. Finally, TCE-M was cooked in boiling water (100 ± 2 °C) for 15–20 min, cooled down and kept in an aluminum-foil bag. The amounts of curcuminoids were determined using a high-performance liquid chromatography-ultraviolet/visible detection (HPLC-UV/Vis) method as will be described below [15]. 

#### 2.2.3. Turmeric Curcumin and Green Tea Extract-Fortified Macaroni 

Turmeric curcumin and green tea extract-enriched macaroni (TCE/GTE-M) was manufactured by mixing GTE-M dough and TCE-M dough, which had been previously described in the Section 2.2.1 and Section 2.2.2, respectively [31], at a ratio of 1:1 (*w*/*w*). It was then manually kneaded for 25 min, cooked in boiling water (100 ± 2 °C) for 15–20 min, cooled down, and kept in an aluminum-foil bag.

### 2.3. Determination of Total Phenolic Content 

Total phenolic content (TPC) was determined using the colorimetric Folin-Ciocaltue method [32]. Macaroni products (1 g) were dissolved in DI water (5 mL), mixed thoroughly using a blender (Waring Laboratory 7010S, 1 L capacity, Electrolux^®^, Stockholm, Sweden) and filtered through a clean cloth sheet. In the assay, macaroni slurry or standard GA (0.1 mL), Folin-Ciocalteu reagent (0.1 mL), and 7% (*w*/*v*) sodium carbonate (1.0 mL) were mixed, incubated at room temperature for 30 min and the optical density (OD) of the product was measured at 765 nm against the reagent blank using a double-beam UV-Vis spectrophotometer (Shimadzu UV-1700 Series, Shimadzu Scientific Instruments, Nakakyo, Kyoto, Japan). The amount of TPC was then calculated from the standard curve of GA (8.5–170 μg/mL) and reported as a value of gallic acid-equivalent (GAE).

### 2.4. HPLC Analysis of Catechins

Catechins in GTE were quantified using the reverse-phase HPLC-UV/Vis detection method [26]. The HPLC system included a column (ODS type, 250 mm × 4.6 mm, 5 μm particle size, temperatures in a range of 25–30 °C), mobile-phase solvents A (0.2% o-phosphoric acid/acetonitrile/THF (86/12.5/1.5, *v*/*v*/*v*), and B (0.2% ortho-phosphoric acid/acetonitrile/THF (73.5/25/1.5, *v*/*v*/*v*) for linear gradient elution, and a UV/Vis detector. The elution protocol was adhered to by adding 0–100% of mobile phase A (30 min), while the volume of mobile phase B was gradually increased from 0–100% (10 min, hold 20 min) at a flow rate of 1.0 mL/min. Mobile phase B was then slowly decreased to an initial concentration of 100% mobile phase A (10 min, hold 20 min). Additionally, the detection wavelength was set at 280 nm, and the duration of the running time was 90 min. Authentic catechins, including EC, EGCG and ECG (0–100 mg/mL), were used for construction of calibration curves, and each catechin concentration was determined from the curves.

### 2.5. HPLC-Based Analysis of Curcuminoid Content

Curcuminoids, including *bis*-desmethoxycurcumin, curcumin and desmethoxycurcumin, were quantified by applying the HPLC system (Shimadzu LC-10ATVP HPLC system, Shimadzu Scientific Instruments, Nakakyo, Kyoto, Japan) connected to a UV-Vis detector (Shimadzu SPD-10 AVVP UV/Vis detector, Shimadzu Scientific Instruments, Nakakyo, Kyoto, Japan) as had been established by Hirun and coworkers [15]. In our analysis, samples (10 old) were injected automatically, fractionated onto a column (Prodigy ODS-3 type, 250 mm × 4.6 mm, 5 μm particle size, Phenomenex Inc., Torrance, CA, USA) and eluted with gradient-mode using the mobile-phase solutions A (1% acetic acid and 99% DI water, *v*/*v*) and B (45% acetonitrile and 54% DI water, *v*/*v*) at a flow rate of 1 mL/min. The process was then allowed to run for 30 min, and the detected curcuminoids were eluted at 425 nm. The amounts of curcuminoids were then determined from each calibration curve of the curcuminoids.

### 2.6. Determination of Antioxidant Activity

Antioxidant activity was determined using the Trolox-equivalent antioxidant capacity (TEAC) method [32]. Working ABTS radical cation (ABTS^●+^) solution was obtained by oxidation of 7 mM ABTS stock solution with 2.45 mM K_2_S_2_O_8_ (1:1, *v*/*v*) and stored at room temperature for 12–16 h. Before use, the working ABTS^●+^ solution was diluted with DI water and photometrically measured at 734 nm to reach an OD of 0.70 ± 0.02. Ten microliters of DI water (blank), Trolox (standard) and macaroni slurry were incubated with the diluted ABTS^●+^ solution in the dark at 25 °C for exactly 6 min. OD values were then measured photometrically at 764 nm against a reagent blank. Results have been reported as mg TEAC/g dry weight.

### 2.7. Determination of Nutritional Facts 

GTE-M, TCE-M and GTE/TCE-M products were randomly selected for analysis of their nutritional fact values at the Central Laboratory (Chiang Mai), Maerim, Chiang Mai, Thailand. Total calories and total carbohydrate contents were determined using the established methods of the International Officers and Committees 1993 [33] with slight modifications [34]. Amounts of dietary fiber and protein were measured according to the methods prescribed by the Official Methods of Analysis, the 21st Edition, AOAC International, Rockville, MD, USA [35] with slight modifications [34].

### 2.8. Determination of Physical Properties 

The color indices including lightness (L*), greenness-redness (a*) and blueness-yellowness (b*) of fortified macaroni products were measured with a CR400 Konica Minolta Chroma Meter (Tokyo, Japan) using the methods established by Hirun et al. [36]. Notably, L* = 0 indicates the darkest, while L* = 100 indicates the brightest. Furthermore, a* value is negative indicated by green, while a* value is positive indicated by red. Lastly, b* value is negative indicated by blue, while b* value is positive indicated by yellow.

### 2.9. Satisfactory Test

Based on the Central Composite Design, scores of macaroni products were evaluated by semi-trained sensory panelists who were familiar with the products using a nine-point hedonic scale test [37]. The panelists were divided into three groups: panel 1 subjects (*n* = 100) received the GTE-M product, panel 2 subjects (*n* = 100) received the TCE-M product and panel 3 subjects (*n* = 200) received the GTE/TCE-M product. Sessions were conducted in an air-conditioned sensory test laboratory under conditions of controlled lighting and ambient temperatures (22–25 °C) at the Division of Product Development Technology, Faculty of Agro-Industry, Chiang Mai University, Chiang Mai, Thailand. In terms of the evaluation process, the macaroni products were blind-tested in triplicate by members of the panels, wherein the order in which each type of macaroni was presented to the participants was random. At each session, the macaroni products were served in transparent plastic plates coded with 3-digital numbers. The subjects were asked to score the degree of intensity in terms of the sensory attributes using a 9-point hedonic scale ranging from 1 = dislike extremely, 2 = dislike very much, 3 = dislike moderately, 4 = dislike slightly, 5 = neither like nor dislike, 6 = like slightly, 7 = like moderately, 8 = like very much to 9 = like extremely.

### 2.10. Assessment of Blood Glucose and Serum Lipid Levels in Rats

#### 2.10.1. Animal Care

The protocol for animal experimentation was approved of by the Ethical Committee for Animal Study at the Faculty of Medicine, Chiang Mai University, Chiang Mai, Thailand (Protocol Number 42/2556). Male wild-type Wistar rats were purchased from a National Animal Center, Mahidol University, Salaya Campus, Nakornpathom, Thailand. They were housed separately in polyethylene cages with free access to feed and clean drinking water under controlled conditions of temperature (20–22 °C), humidity (50 ± 10%) and light (12-h light/dark cycle). Normal diets consisted of 62.5% carbohydrates, 2.5% fats (0.11% ω3-polyunsaturated fatty acid, 0.53% monounsaturated fatty acid and 0.52% saturated fatty acid), 14.3% protein and 3.7% fiber by weight and total energy of 100 kcal/g. Staple diet ingredients were purchased from Charoen Pokaphan Food Public Company Limited, Bangkok, Thailand and used throughout the course of this study.

#### 2.10.2. Intervention of Streptozotocin-Induced Diabetic Rats

Rats with body weights (BW) in a range of 170–300 g were randomly divided into nine groups (*n* = 6 each). All groups received equal treatment. However, group 3 (*n* = 4) received a single intraperitoneal (i.p.) injection of normal saline solution (NSS) as a control group or received a freshly prepared STZ solution (65 mg/kg BW) dissolved in NSS immediately before being used (1 mL/kg) in order to induce type-2 diabetic mellitus according to previously established methods [38,39]. A drop of fasting (8–12 h) blood was collected from the tail veins of rats to estimate fasting blood glucose (FBG) levels after two days using a Blood Glucose Monitor (ONETOUCH^®^ SelectSimple™, Johnson & Johnson Limited Company, New Brunswick, NJ, USA). Only animals with FBG levels ≥ 180 mg/dL were selected for this study. Macaroni products were ground using an electrical blender before being used. Diabetic rats were randomly divided into eight groups (*n* = 6 each) and administered per oral (p.o.) with DI (0.5 mL/kg), glibenclamide (5 mg/kg), GTE-M (30 and 300 mg/kg), TCE-M (30 and 300 mg/kg) and GTE/TCE-M (30 and 300 mg/kg) once a day in the morning for four weeks. Rats that had been injected with NSS (normal group, N) were administered p.o. with DI. All rats were fed a normal diet and given access to clean drinking water ad libitum throughout the course of the study. BW values were recorded weekly. On weeks 2 and 4 of the study, the treated rats were fasted overnight for 8–12 h. Blood samples (1 mL) were then withdrawn from their tail veins and collected in plain vacutainer tubes. Clotted blood was centrifuged at 2000 g for 10 min, and the serum was separated for measurement of FBG, triglyceride (TG) and total cholesterol (TC) concentrations using the methods described below.

#### 2.10.3. Laboratory Investigation 

FBG, TG and TC concentrations in the serum were measured using an automated Chemistry Analyzer (Cobas^®^ Series 4000, Roche Diagnostics, Indianapolis, IN, USA) according to the manufacturer’s instructions at the Investigation Laboratory, Animal Hospital, Faculty of Veterinary Medicine, Chiang Mai University, Chiang Mai, Thailand. 

### 2.11. Statistical Analysis

Results were analyzed using the SPSS Statistics Program (IBM SPSS^®^ Software version 22, IBM Corporation, Armonk, NY, USA, shared license by Chiang Mai University, Thailand). Data are expressed as mean ± standard deviation (SD) values. Descriptive analysis for the results of the sensory acceptance assessment was employed to describe all variables related to the macaroni products and sensory analysis. Statistical significance was determined using one-way analysis of variance (ANOVA) followed by Tukey’s HSD post-hoc test, and statistical significance was taken at a *p* value < 0.05.

## 3. Results and Discussion

### 3.1. Chemical Composition of Macaroni Products

According to our analysis, turmeric was abundant with phenolics (TPC = 56.92 ± 6.96 mg GAE/g fresh weight and 46.68 ± 8.88 mg GAE/g dry weight) and curcuminoids (19.39 ± 3.38 mg/g fresh weight and 19.04 ± 0.38 mg/g dry weight). Additionally, *Camellia sinensis* tea leaves were found to be rich in phenolics (TPC = 82.31 ± 8.92 mg GAE/g fresh weight and 72.67 ± 32.40 mg GAE/g dry weight), which were attributed to EC (TPC = 16.32 ± 0.54 mg/g fresh weight and 16.52 ± 0.39 mg/g dry weight), EGCG (TPC = 17.56 ± 0.49 mg/g fresh weight and 16.44 ± 4.18 mg/g dry weight) and ECG (TPC = 0.82 ± 0.21 mg/g fresh weight and 1.03 ± 0.27 mg/g dry weight). According to the recipe and formulation described above, consumable macaroni products including GTE-M, TCE-M and GTE/TCE-M (Figure 1A–C) were manufactured and made available for the experiments. 

As is shown in Table 1, GTE/TCE-M exhibited stronger antioxidant activity than GTE-M and TCE-M (*p* < 0.05), of which GTE-M was more potent than TCE-M. In addition, TPC was highest in GTE-M, for which GTE/TCE-M contained more TPC than TCE-M. Likewise, different amounts of EC, EGCG, ECG and curcuminoids were detected in these macaroni products. The results suggest that curcuminoids and phenolics, including catechins, contribute to antioxidant activity. 

We previously used the microwave heating technique to prepare EGCG-abundant GTE for therapeutic purposes ROS-scavenging, anti-lipid peroxidation and iron-chelating properties under iron-overloaded conditions [26,40,41,42,43]. Green tea contains high amounts of catechins [2,3], for which the abundancy order was EGCG > ECG > EGC > EC [4] and the antioxidant potency order was ECG > EGCG > EGC > GA > EC = C [3]. In addition, we have reported on the antioxidant, free radical-scavenging and anti-lipid peroxidation activities exhibited by curcuminoids in animals and humans [21,44,45].

The nutritional facts of the three macaroni products are shown in Table 2. In our analysis, total energy and carbohydrate contents were not found to be significantly different among the three products, whereas protein, moisture and ash contents in the GTE-M product were significantly higher than those in the TCE-M and GTE/TCE-M products. On the other hand, the GTE-M product contained significantly lower fat and crude fiber contents than the other two products. 

Surprisingly, Olfat and colleagues revealed that the color, flavor and appearance of semolina macaroni were improved by adding whey carboxymethyl cellulose-protein rather than corn hydroxyethyl cellulose protein [31]. Fortification of macaroni made from durum wheat semolina was then combined with green tea and turmeric extracts to mask the undesirable astringent and bitter taste of the existing tannins. This also offset the spicy taste of curcuminoids and enriched the antioxidant and antihyperlipidemic effects of the phenolic compounds. Similarly, alginate microencapsulation of Azolla fern powder and fortification in macaroni were purposed to mask the undesirable taste and odor of Azolla, as well as to minimize any loss in antioxidant activity. However, the overall acceptability index scores of the encapsulated macaroni were not affected by these modifications [27]. Interestingly, the enrichment of the macaroni with multiple herb and spice vegetable blends led to an improvement in flavor variety and increased the intake of the meal among participants [30].

As shown in Table 3, fortification of macaroni increased brightness (L*) values but decreased the greenness (a*) values of the products significantly when compared with GTE fortification, whereas the two parameters between the TCE-M and the GTE/TCE-M were not found to be significantly different. Moreover, fortification of macaroni with TCE resulted in a significantly greater yellowness (b*) value than fortification with GTE or GTE/TCE, whereas the yellowness values of GTE-M and GTE/TCE-M were not observed to be significantly increased. Thus, fortification effectively influenced the color parameters. Microwave-assisted heating is an important step in the manufacture of green tea to inactivate polyphenol oxidase in tea leaves. Accordingly, green tea contains the highest phenolic and catechin contents, as well as presenting a bright color, sweet taste and pleasant odor [46]. Additionally, microwave-exposed GTE was fortified in bread, catechins; particularly, EGCG and EGC were stable for four days at room temperature and up to nine weeks at −20 °C [47]. Likewise, noodles incorporated with GTE ingredients were found to show higher surface lightness due to a loss in chlorophyll content [48]. Curcumin is a curcuminoid pigment responsible for the yellow color of turmeric and may affect the original color of macaroni as a food ingredient. The addition of curcumin microcrystals significantly decreased levels of lipid peroxidation; nonetheless, this addition can have an effect on the product’s color and decreased the degree of acceptance of the product in terms of consumer preference [49]. Thus, proper utilization of tea polyphenols and turmeric curcuminoids in food products can be beneficial and improve the antioxidant capacity and nutritional quality of food. However, it would be of significant interest to further investigate the color stability and shelf-life of green tea- and turmeric curcumin-fortified macaroni products.

The scores pertaining to consumer acceptance were evaluated using a nine-point hedonic scale test and the results are shown in Table 4. Interestingly, subjects in panel 1 expressed their preference for the GTE-M products with responses ranging from “dislike slightly” to “neither like nor dislike” (range of 4.7–5.1), while subjects in panel 2 expressed their preference for TCE-M products with responses ranging from “neither like nor dislike” to “like slightly” (range of 5.9–6.7). Surprisingly, subjects in panel 3 expressed their preference for GTE/TCE-M products with responses ranging from “like slightly” to “like very much” (range of 6.2–8.2).

### 3.2. Effects of Consumption of Macaroni on Diabetic STZ-Induced Rats

According to our findings, FBG levels of STZ-induced rats significantly rose over 1–5 weeks when compared with NSS-induced rats indicating type 2-DM status, while the increasing values of FBS were significantly decreased by treatments involving the antidiabetic drug glibenclamide (5 mg/kg/day, p.o.) in week 5 (Figure 2A). Not surprisingly, consumption of the GTE-M product (30 and 300 mg/kg/day) did not influence FBG levels of diabetic rats over the five weeks of this study, during which the level of GTE-M (30 mg/kg/day) did seem to slightly increase FBG levels (Figure 2B). Similarly, the consumption of TCE-M and GTE/TCE-M (30 and 300 mg/kg/day) did not influence FBG levels (Figure 2C,D, respectively). Although all the macaroni products failed to restore high blood glucose levels in STZ-induced diabetic rats, they did not further enhance the increased blood glucose levels. 

Similarly, serum TG levels of STZ-induced rats were increased in week 1 (*p* < 0.05) and in weeks 3 and 5 when compared with NSS-treated rats. The increased serum TG levels were not restored by glibenclamide treatment throughout the course of this study (Figure 3A). Consumption of GTE-M (30 mg/kg) tended to lower the increased serum TG levels over weeks 3–5 in diabetic rats when compared with rats of the DI-treated group; in contrast, consumption of GTE-M (300 mg/kg) slightly enhanced the increasing serum TG levels (Figure 3B). Likewise, the feeding of TCE-M and GTE/TCE-M products were found to slightly enhance the increasing TG levels of diabetic rats when compared with the DI-fed rats (Figure 3C,D).

Consistently, serum TC levels were significantly increased in STZ-induced rats, especially in the first week of the study, when compared with NSS-fed rats. The increased TC levels were then significantly decreased after glibenclamide treatment (Figure 4A). Surprisingly, consumption of GTE-M (30 mg/kg), but not GTE-M (30 mg/kg), tended to lower the increased serum TC levels of diabetic STZ-induced rats during the first week when compared to rats fed with DI (Figure 4B). Nonetheless, feeding rats with TCE-M and GTE/TCE-M products for five weeks did not influence serum TC levels of diabetic STZ-induced rats when compared with rats fed with DI (Figure 4C,D, respectively).

Foods that are low on the glycemic index (GI) scale tend to release glucose slowly and steadily, whereas foods high on the GI scale release glucose rapidly. Interestingly, consumption of macaroni provided significantly greater GI (68 ± 8) contents than did consumption of spaghetti (45 ± 6) [50]. Since macaroni has a moderate GI value, it has been suggested for and offered to patients with diabetes or obesity [51]. Moreover, complex carbohydrates are slowly digested by digestive enzymes in the small intestines, producing GI values in the order that follows: white bread > whole wheat bread > macaroni > tarhana soup > white rice > potatoes > noodle soup [52]. Additionally, curcumin and bis-desmethoxycurcumin presented antioxidant, anti-lipid peroxidation, anti-hyperlipidemic and hepatoprotective effects against nicotine-induced lung toxicity in Wistar rats [53,54]. Herein, we have reported that the increased levels of serum total cholesterol in diabetic STZ-induced rats were tentatively lowered by consumption of GTE-M and GTE/TCE-E products, which was similar to the outcomes of the glibenclamide treatment. However, increased TG levels were not necessarily decreased by all the treatments. The results suggest that the GTE- and GTE/TCE-fortified macaroni should be given to diabetic patients in order to mitigate oxidative stress and hypercholesterolemia complications.

## 4. Conclusions

Three macaroni products were prepared from semolina and were then fortified with green tea, turmeric and the two extracts, of which the GTE-M product was most abundant with phenolics and the GTE/TCE-M product possessed the most potent antioxidation properties. The consumer panel expressed the greatest preference for the GTE/TCE-M product. Neither of the other two products exacerbated hyperglycemia in STZ-induced rats, while the GTE-M product tended to lower the increased serum cholesterol levels. Altogether, the incorporation of green tea extract along with turmeric extract resulted in the best compromise in terms of the technological, chemical, nutritional, sensorial and metabolic aspects of enriched macaroni. In conclusion, macaroni enriched in this manner should help to provide consumers with antioxidant and lipid-lowering formulations of green tea and turmeric that would be beneficial for health. Furthermore, the contributions of green tea and turmeric to functional macaroni will need to be clinically investigated in patients with diabetes and hypercholesterolemia. 

## Figures and Tables

**Figure 1 foods-12-00534-f001:**
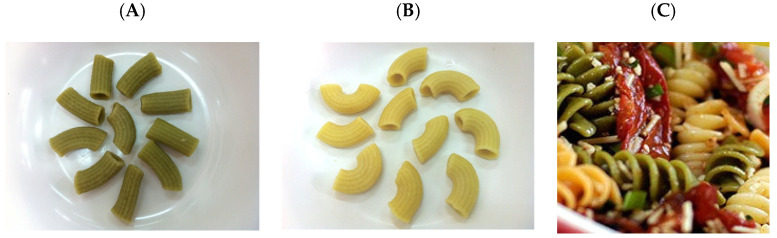
Cooked green tea extract-fortified macaroni (**A**), turmeric curcumin extract-fortified macaroni (**B**) and green tea extract/turmeric curcumin-fortified macaroni (**C**) products.

**Figure 2 foods-12-00534-f002:**
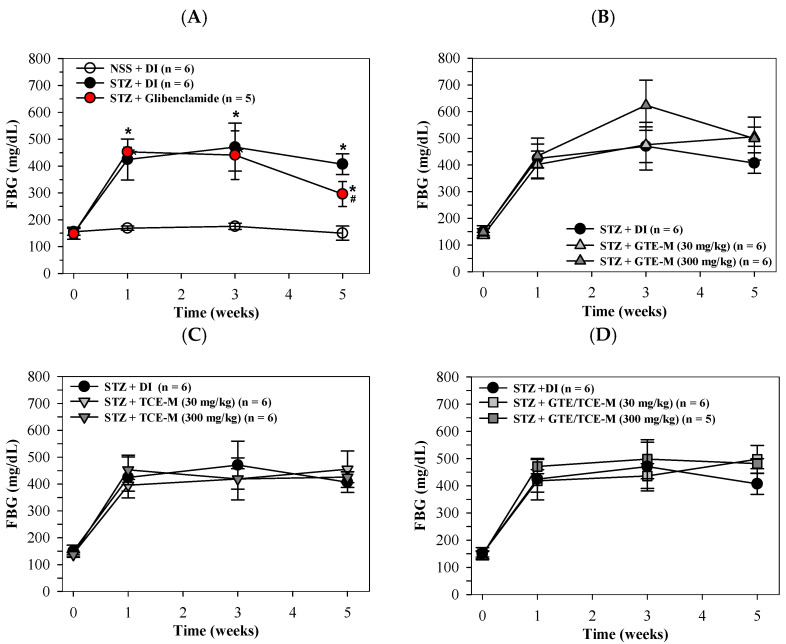
Serum levels of FBG in STZ-induced rats intervened with macaroni products. Rats were i.p. injected with STZ (65 mg/kg, single dose) to induce diabetic mellitus and administered p.o. with DI, glibenclamide (5 mg/kg) (**A**), GTE-M (**B**), TCE-M (**C**) or GTE/TCE-M (**D**) macaroni products (30 and 300 mg/kg each) every day for four weeks. Data are expressed as mean ± SD values. Accordingly, * *p* < 0.05 when compared with NSS + DI at the same time; while ^#^
*p* < 0.05 when compared with STZ + DI. Abbreviations: DI = deionized water, FBG = fasting blood glucose, GTE-M = green tea extract-fortified macaroni, GTE/TCE-M = green tea extract and turmeric curcumin extract-fortified macaroni, i.p. = intraperitoneally, NSS = normal saline solution, p.o. = per oral, STZ = streptozotocin and TCE-M = turmeric curcumin extract-fortified macaroni.

**Figure 3 foods-12-00534-f003:**
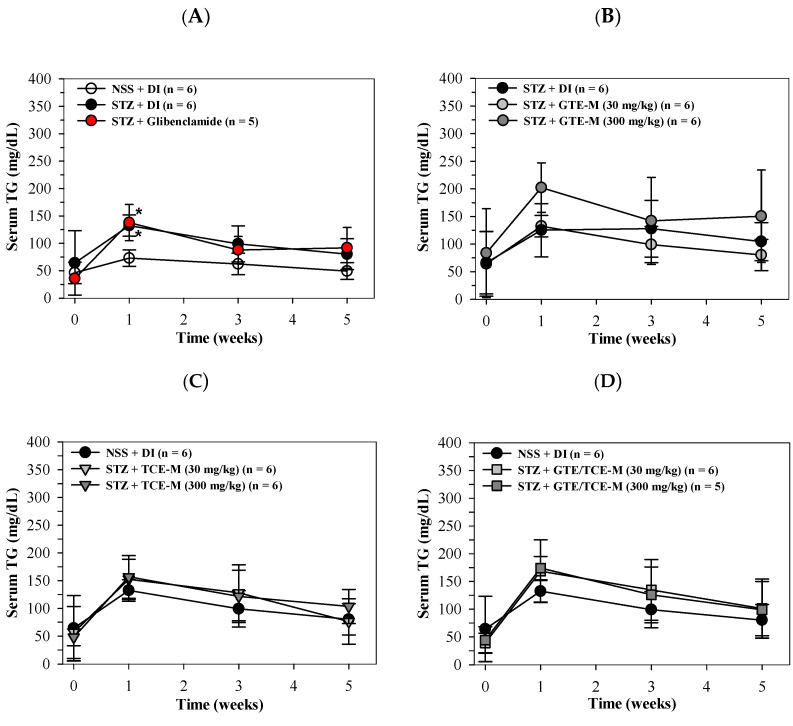
Serum levels of triglyceride in STZ-induced rats intervened with macaroni products. Wistar rats were intraperitoneally injected with STZ (65 mg/kg, single dose) to induce diabetic mellitus or NSS (control) and orally administered with DI, glibenclamide (5 mg/kg) (**A**), GTE-M (**B**), TCE-M (**C**) or GTE/TCE-M (**D**) macaroni products (30 and 300 mg/kg each) every day for four weeks. Data are expressed as mean ± SD values. Accordingly, * *p* < 0.05 when compared with NSS + DI at the same time. Abbreviations: DI = deionized water, GTE-M = green tea extract-fortified macaroni, GTE/TCE-M = green tea extract and turmeric curcumin extract-fortified macaroni, NSS = normal saline solution, STZ = streptozotocin, TCE-M = turmeric curcumin extract-fortified macaroni and TG = triglyceride.

**Figure 4 foods-12-00534-f004:**
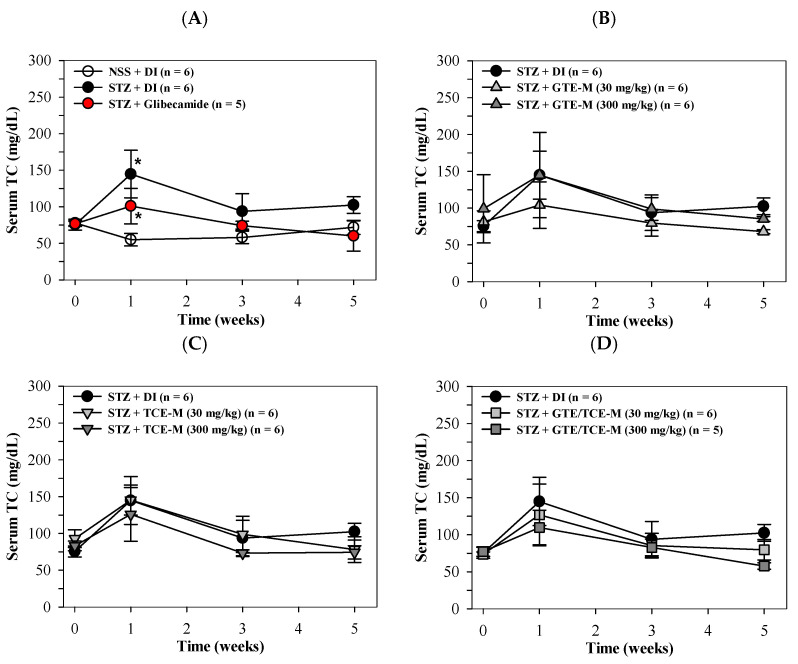
Total cholesterol (TC) levels of serum in STZ-induced rats. Wistar rats were intraperitoneally injected with STZ (65 mg/kg, single dose) to induce diabetic mellitus or NSS (control) and orally administered with DI, glibenclamide (5 mg/kg) (**A**), GTE-M (**B**), TCE-M(**C**) or GTE/TCE (**D**) macaroni products (30 and 300 mg/kg each) every day for four weeks. Data are expressed as mean ± SD values. Accordingly, * *p* < 0.05 when compared with NSS + DI at the same time interval. Abbreviations: DI = deionized water, GTE-M = green tea extract-fortified macaroni, GTE/TCE-M = green tea extract and turmeric curcumin extract-fortified macaroni, NSS = normal saline solution, STZ = streptozotocin, TC = total cholesterol, and TCE-M = turmeric curcumin extract-fortified macaroni.

**Table 1 foods-12-00534-t001:** Antioxidant activity, TPC, catechin and curcuminoid contents were all present in the three macaroni products.

Analysis	Macaroni Products		
	GTE-M	TCE-M	GTE/TCE-M
Antioxidant activity (mg TEAC/g)	7.07 ± 0.29 ^a^	1.61 ± 0.09	12.52 ± 0.92 ^a,b^
TPC (mg GAE/g)	10.32 ± 0.09 ^a^	4.53 ± 0.16 ^c^	6.82 ± 0.03 ^b^
EGCG (mg/g)	2.74	ND	1.37
EC (mg/g)	2.47	ND	1.24
ECG (mg/g)	0.17	ND	0.09
Curcuminoids (mg/g)	ND	0.96	0.48

Data on antioxidant activity and TPC were obtained from three independent repetitions and are expressed as mean ± SD values. The amounts of catechins and curcuminoids are presented as average values. Within the same parameters, the values followed by different superscript letters (^a,b,c^) are indicative of significant differences (*p* < 0.05). Abbreviations: EGCG = epigallocatechin-3-gallate, GAE = gallic acid equivalent, GTE-M = green tea extract-fortified macaroni, GTE/TCE-M = green tea extract and turmeric curcumin extract-fortified macaroni, ND = not determined, TCE-M = turmeric curcumin extract-fortified macaroni, TEAC = Trolox-equivalent antioxidant capacity, TPC = total phenolic content.

**Table 2 foods-12-00534-t002:** Nutritional facts of three macaroni products. Data obtained from three independent repetitions are expressed as mean ± SD values. Within the same parameters, values followed by different superscript letters (^a,b^) are indicative of significant differences (*p* < 0.05).

Analytes	Macaroni Products		
	GTE-M	TCE-M	GTE/TCE-M
Total energy (kcal/g)	3.29 ± 0.13 ^b^	3.15 ± 0.02 ^b^	3.59 ± 0.04 ^a^
Carbohydrate (%)	78.33 ± 0.13	79.02 ± 0.57	78.50 ± 0.01
Protein (%)	10.73 ± 0.15 ^a^	9.41 ± 0.58 ^b^	9.50 ± 0.04 ^b^
Fats (%)	1.00 ± 0.18 ^b^	1.79 ± 0.06 ^a^	1.99 ± 0.04 ^a^
Moisture (%)	4.19 ± 0.16 ^a^	3.72 ± 0.06 ^b^	3.98 ± 0.21 ^a,b^
Crude fiber (%)	3.63 ± 0.26 ^b^	4.16 ± 0.08 ^a^	4.12 ± 0.04 ^a,b^
Ash (%)	2.14 ± 0.11 ^a^	1.91 ± 0.04 ^b^	1.92 ± 0.04 ^a,b^

Abbreviations: GTE-M = green tea extract-fortified macaroni, GTE/TCE-M = green tea extract and turmeric curcumin extract-fortified macaroni, TCE-M = turmeric curcumin extract-fortified macaroni.

**Table 3 foods-12-00534-t003:** Physical properties of cooked GTE-M, TCE-M and GTE/TCE-M products. Data obtained from triplicate experiments are expressed as mean ± SD values. Within the same column, values presented with different superscript letters (^a,b^) are significantly different (*p* < 0.05) from each other.

Macaroni Product	L*	a*	b*
GTE-M	49.91 ± 1.55 ^b^	−1.43 ± 0.16 ^a^	13.87 ± 0.93 ^b^
TCE-M	59.80 ± 1.12 ^a^	−3.91 ± 0.33 ^b^	20.86 ± 0.88 ^a^
GTE/TCE-M	56.09 ± 3.38 ^a^	−3.90 ± 1.19 ^b^	17.06 ± 2.69 ^b^

Abbreviations: a* = greenness-redness, b* = blueness-yellowness, GTE-M = green tea extract-fortified macaroni, GTE/TCE-M = green tea extract and turmeric curcumin extract-fortified macaroni, L* lightness, TCM = turmeric curcumin extract-fortified macaroni.

**Table 4 foods-12-00534-t004:** Sensory test with respect to GTE-M, TCE-M and GTE/TCE-M products using a 9-point hedonic scale. A 9-point hedonic scale was applied as follows: 1 = dislike extremely, 2 = dislike very much, 3 = dislike moderately, 4 = dislike slightly, 5 = neither like nor dislike, 6 = like slightly, 7 = like moderately, 8 = like very much to 9 = like extremely. Data are expressed as mean ± SD values. Within the same parameters, values followed by different superscript letters (^a,b^) are indicative of significant differences (*p* < 0.05).

Characteristics	Sensory Evaluation Score		
	GTE-M (*n* = 100)	TCE-M (*n* = 100)	GTE/TCE-M (*n* = 200)
Appearance	4.9 ± 1.3 ^b^	6.7 ± 0.7 ^a,b^	7.7 ± 1.1 ^a^
Green colour	4.7 ± 1.2 ^ns^	ND	6.7 ± 1.2 ^ns^
Yellow colour	ND	6.7 ± 0.8 ^ns^	7.0 ± 1.2 ^ns^
Green tea aroma	5.4 ± 1.6 ^ns^	ND	6.7 ± 1.2 ^ns^
Curcumin aroma	ND	5.9 ± 0.9 ^ns^	6.8 ± 1.1 ^ns^
Total aroma	5.4 ± 1.4 ^b^	6.1 ± 0.9 ^a,b^	7.1 ± 1.1 ^a^
Total taste	5.3 ± 1.2 ^b^	6.2 ± 1.1 ^a,b^	7.2 ± 1.0 ^a^
Texture	5.3 ± 1.0 ^b^	6.0 ± 1.1 ^a,b^	7.2 ± 0.9 ^a^
Overall preferences	5.2 ± 0.8 ^b^	6.1 ± 0.9 ^a,b^	7.2 ± 1.0 ^a^

Abbreviations: GTE-M = green tea extract-fortified macaroni, GTE/TCE-M = green tea extract and turmeric curcumin extract-fortified macaroni, ND = not determined, ns = not significant, TCE-M = turmeric curcumin extract-fortified macaroni.

## Data Availability

The data presented in this study are available on request from the corresponding authors.

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
