# Peer review of "Testing the Feasibility and Dietary Impact of Macaroni Fortified with Green Tea and Turmeric Curcumin Extract in Diabetic Rats"

_foods, 2023, doi:10.3390/foods12030534_

Round 1
Reviewer 1 Report
Regarding MS entitled ‘’ Chemical, nutritional, sensory and antidiabetic evaluation of macaroni fortified with green tea and turmeric curcumin in rats ‘’
L15. Macaroni can be considered a promising healthy and functional food, if Macaroni is healthy functional food why the authors used green tea and turmeric curcumin fortified macaroni? It seems confusing to the readers.
L21. The experimental design is missing in the abstract. Please add details about the design before results.
L22. What do you mean by performance here?
L24. Three preparation and the authors mentioned only two!!
The conclusion of the abstract is confusing please be precise and add strong recommendation to the readers.
L252. Nine groups (n=6), please revise and write down the total number of rats
Statistical analysis why some data are expressed as SD and other as SEM. Also, why the authors used T test for which data. Please define in this section. I didn’t see any data presented as SEM and also for T test.
Regarding figures why the NSS = normal saline solution was not included in the statistical analysis? It should be included in the statistical analysis to compare.
L435. greater antioxidant activity, in the current study the authors did not measure any of the antioxidant markers either in blood or in rat tissues. Please revise.
L466-472. The authors did not measure these parameters!!
L487. oxidative stress again is not measured in the study. The study focused on blood glucose and lipid profile.
Reviewer 2 Report
Manuscript recieved for review new type of macaroni product is prepared and proposed, which was supplemented with green tea and turmeric curcumin extracts. Authors investigated new products’ chemical compositions and antioxidant activity, the degree of consumer satifaction, anti−diabetic and anti−hyperlipidemic effects on streptozotocin−induced and high−fat diet fed rats.
Applied analysis to the new type of macaroni product is enough elaborate and comprehensive for good quality research.
Title of the manuscript is unclear and needs rephrasing.
Intorduction section is adequate and elatorate. The aim of the study is detailed.
Material and methodes section is appropriate for the described and conducted testing.
Results and discussion are divided in two sections.
Although authors' instructions propose the manuscript structure of results and discussion as separate sections, it is strongly suggested to combine these two sections in one, integral section. In that way it would be much easier to follow presented research.
The quality of presented results is very good, with very elaborate discussion and referencting to other authors’ results.
Conclusion section is appropriate for the presented results.
More needed correctoins are noted in manuscripts’ pdf file.
Decission: minor revision

Reviewer 3 Report
Chemical, nutritional, sensory and antidiabetic evaluation of macaroni fortified with green tea and turmeric curcumin in rats. The manuscript is very well written and contributes to the field. In my opinion, this manuscript need addressing the suggestions below.
Lines 15-19: please revise these lines and focus on background of the study instead of saying benefits of
Introduction is appropriate and provided with support literature.
Sections 2.2.1., 2.2.2., and 2.2.3: Please provide citation
Color (L, a, and B) characteristics of the samples must be performed. Fortification may significantly change the color parameters. Authors must consider performing color
Lines 163-166: very long sentence
Methodology is appropriate and provided with supported literature.
Figure 2. Improve the quality of figures
Figure 3. Improve the quality of figures
Figure 4. Improve the quality of figures
Line 417: did not differ can be revise as showed insignificant difference
Discussion is extremely poor and authors should discuss available literature and cite related studies.
Figure 1 C. (C) products… WHAT DOES IT MEAN?
Line 538: in vitro should be in Italics
Line 562: Curcumin/Turmeric should be curcumin/turmeric
Lines 616-617: check this
645-646: check this
Round 2
Reviewer 1 Report
Dear authors,
Thank you for your revisions.
Author Response
Thank you so much.